# Most frequent South Asian haplotypes of *ACE2* share identity by descent with East Eurasian populations

**Anshika Srivastava[1], Rudra Kumar Pandey[1], Prajjval Pratap Singh[1], Pramod Kumar[2], Avinash Arvind Rasalkar[3], Rakesh Tamang[4], George van Driem[5], Pankaj Shrivastava[6], Gyaneshwer Chaubey[1]\***

**1** Department of Zoology, Cytogenetics Laboratory, Banaras Hindu University, Varanasi, India, **2** National Centre for Disease Control, Delhi, India, **3** Redcliffe Life Sciences Pvt Ltd, Electronic City, Noida, Uttar Pradesh, India, **4** Department of Zoology, University of Calcutta, Kolkata, India, **5** Institut für Sprachwissenschaft, Universität Bern, Bern, Switzerland, **6** Department of Home (Police), DNA Fingerprinting Unit, State Forensic Science Laboratory, Government of MP, Sagar, India

\* gyaneshwer.chaubey@bhu.ac.in

## Abstract

It was shown that the human Angiotensin-converting enzyme 2 (ACE2) is the receptor of recent coronavirus SARS-CoV-2, and variation in this gene may affect the susceptibility of a population. Therefore, we have analysed the sequence data of *ACE2* among 393 samples worldwide, focusing on South Asia. Genetically, South Asians are more related to West Eurasian populations rather than to East Eurasians. In the present analyses of *ACE2*, we observed that the majority of South Asian haplotypes are closer to East Eurasians rather than to West Eurasians. The phylogenetic analysis suggested that the South Asian haplotypes shared with East Eurasians involved two unique event polymorphisms (rs4646120 and rs2285666). In contrast with the European/American populations, both of the SNPs have largely similar frequencies for East Eurasians and South Asians, Therefore, it is likely that among the South Asians, host susceptibility to the novel coronavirus SARS-CoV-2 will be more similar to that of East Eurasians rather than to that of Europeans.

## Introduction

The novel coronavirus SARS-CoV-2, the causative agent of the ongoing pandemic of COVID-19, today presents one of the major challenges to humanity [1]. Recent studies have effectively demonstrated that the Angiotensin-converting enzyme 2 (ACE2) encoded by a gene located on the X-chromosome is the host receptor for the virus [1, 2]. A decreased level of *ACE2* expression mitigates the severity of the disease. The over-expression or a unique genetic polymorphism of the receptor among Asians have been ruled out in a recent study [3, 4]. ACE2 also maintains cardiovascular homeostasis and electrolyte balance and protects against lung injury by acid aspiration [5]. A comprehensive understanding of *ACE2* variations among various ethnic groups has hitherto been largely unknown.

**Data Availability Statement:** The used data is available at https://evolbio.ut.ee/CGgenomes.html.

**Funding:** This work is supported by the National Geographic Explorer grant HJ3-182R-18. Redcliffe

Life Sciences Pvt Ltd. India provided support in the form of salaries for author AR. The specific roles of this author is articulated in the 'author contributions' section. The funders had no role in study design, data collection and analysis, decision to publish or preparation of the manuscript.

**Competing interests:** One of our author AR is full time employee of Redcliffe Life Sciences Pvt Ltd. India. This does not alter our adherence to PLOS ONE policies on sharing data and materials.

The South Asia subcontinent harbours diverse and endogamous ethnic groups [6]. Most of the genomes of South Asia are autochthonous but show a considerable amount of sharing with East and West Eurasia [7]. However, when we compare overall genome sharing with East *vs.* West Eurasia, South Asians show greater genetic affinity with West Eurasia [8–10]. The only exception is Tibeto-Burman speaking populations, who share a large amount of ancestry with East Eurasia [11]. The genetic structure of *ACE2* haplotypes among South Asian populations is not known. Therefore, we have analysed the whole genome data of South Asians with respect to various world populations for *ACE2* published elsewhere [12, 13] (S1 Table).

## Materials and methods

The research has been approved by the Institutional Ethical Committee of Banaras Hindu University, Varanasi, India. To analyse the *ACE2* among various populations, we have extracted the sequences from the published datasets [12, 13], by using PLINK 1.9 [14]. It has been shown that the 1000 genome dataset for South Asia does not capture the complete South Asian variation, mainly due to unsampled Austroasiatic populations [15]. Hence, we analysed Pagani et al. [12] by way of primary data and further confirmed the results with the 1000 genome data [13]. We extracted 447 samples designated as a diversity set panel in the Pagani et al. data [12]. After excluding samples from Africa, Sahul and relatives up to the second degree, we used 393 samples in all our analyses (S1 Table). A total of 248 polymorphisms were observed in the Pagani et al. data [12] (S2 Table). LD maps for each of the groups were analysed from Haploview [16] (S1 Fig). For both of the datasets, we converted plink file to fasta file (ped to IUPAC) from customised script. Phasing of the data, the calculation of population-wise genetic distances, and Arlequin and Network input files were generated by DnaSP v 6 [17]. The neighbour joining (NJ) tree was constructed by MEGA-X [18] (Fig 1A). Nei's genetic distances and pairwise differences were calculated from Arlequin 3.5 [19] and plotted by R v 3.1 [20] (Fig 1B and S2 Fig). Network v5 [21] and Network publisher were used to construct the median joining (MJ) networks (Fig 2 and S3 Fig). The spatial map of rs4646120 and rs2285666 were drawn from *PGG* toolkit (S4 Fig) [22].

## Result and discussion

Our pooled data have yielded 248 high quality polymorphisms (S2 Table). In the LD (linkage disequilibrium) plot analysis, significant LD blocks of different sizes were present among Caucasus, Central Asians, South Asians, mainland Southeast Asians, insular Southeast Asians and Siberians (S1 Fig). Europeans showed the lowest level of LD. We have used a haplotype based approach for the comparison. In contrast with the genome-wide analysis [8–10], the NJ (Neighbour Joining) tree based on *F*st distances clustered South Asians together with insular and mainland Southeast Asian populations (Fig 1A). This unexpected result suggested closer a genetic affinity of South Asians with East Eurasians for *ACE2*. The pairwise difference analysis suggested lower diversity for South Asian, Southeast Asian and Siberian populations (Fig 1B). Similarly, the 1000 genome populations showed the lowest diversity for East Asian populations (S2 Fig).

The phylogenetic analysis of various haplotypes among studied populations helped to identify the SNPs responsible for the affinity of South Asians with East Eurasians (Fig 2 and S3 Fig). Three major distinct haplotypes were observed. Haplotype 1 (ht1) was more common in West Eurasians, including Central Asian populations, whereas haplotype 2 (ht2) was frequent among East Eurasians, South Asians and Americans (Fig 2 and S3 Fig). Haplotype 3 (ht3) was harboured mainly by East Eurasians and South Asians. The haplotype 2 (ht2) originated from SNP rs4646120, whereas ht3 was derived from SNP rs2285666. Phylogenetically both of these

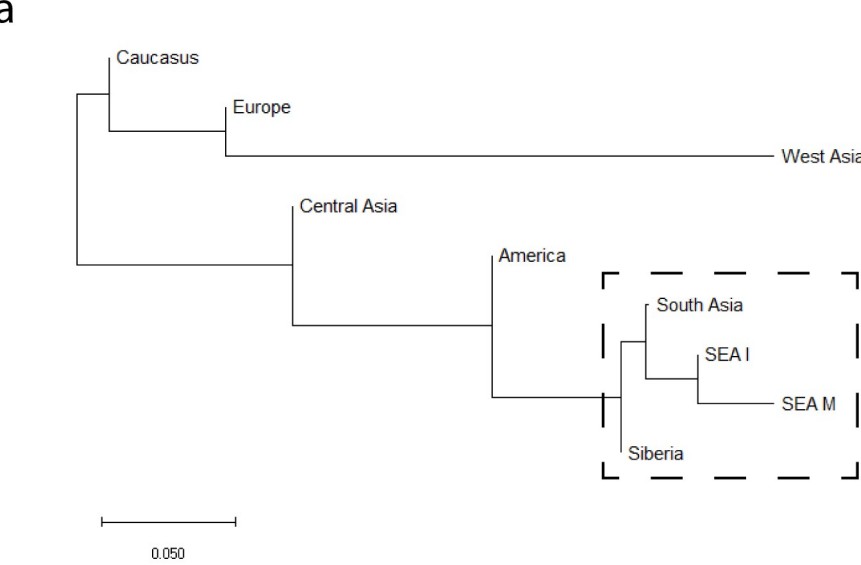

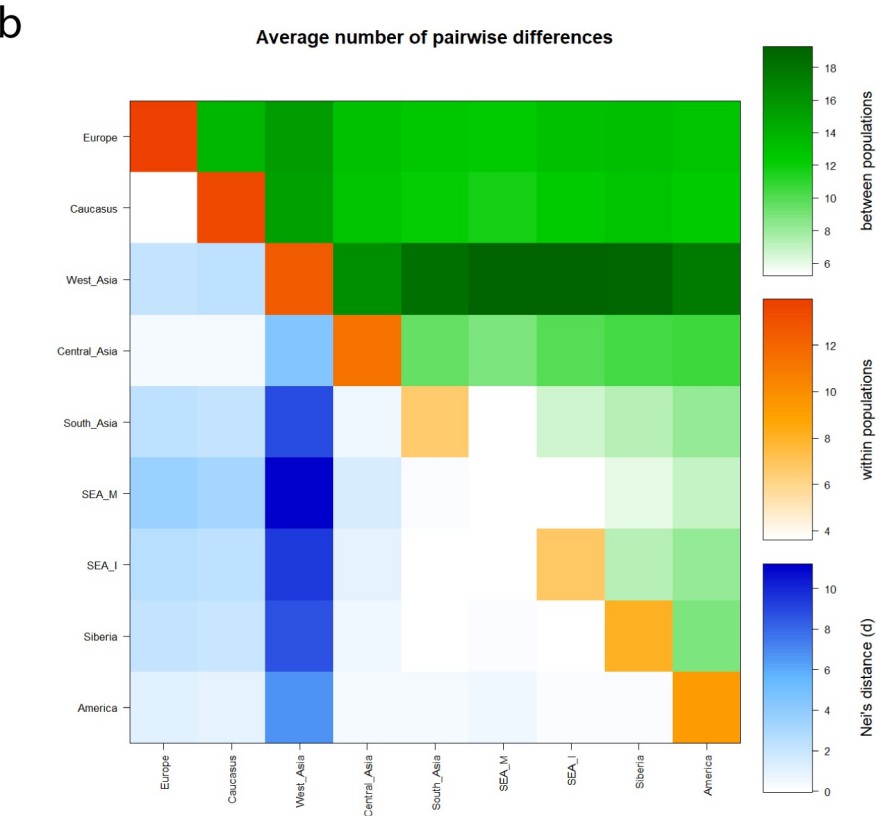

**Fig 1.** a) The Neighbour-Joining (NJ) tree showing the genetic relationship of the studied populations. The figure was drawn from the *F*st distances obtained from the haplotype analysis of Eurasian populations. b) Heat map showing the intra- and inter-population variation measured by average pairwise sequence differences of the ACE2 gene. The average pairwise differences between populations are shown in the upper triangle of the matrix (green). The average number of pairwise differences within each population group are shown along the diagonal (orange). The differences between populations based on Nei's genetic distances are depicted in lower triangle of the matrix (blue). The obtained values of various parameters have been shown at the color scales.

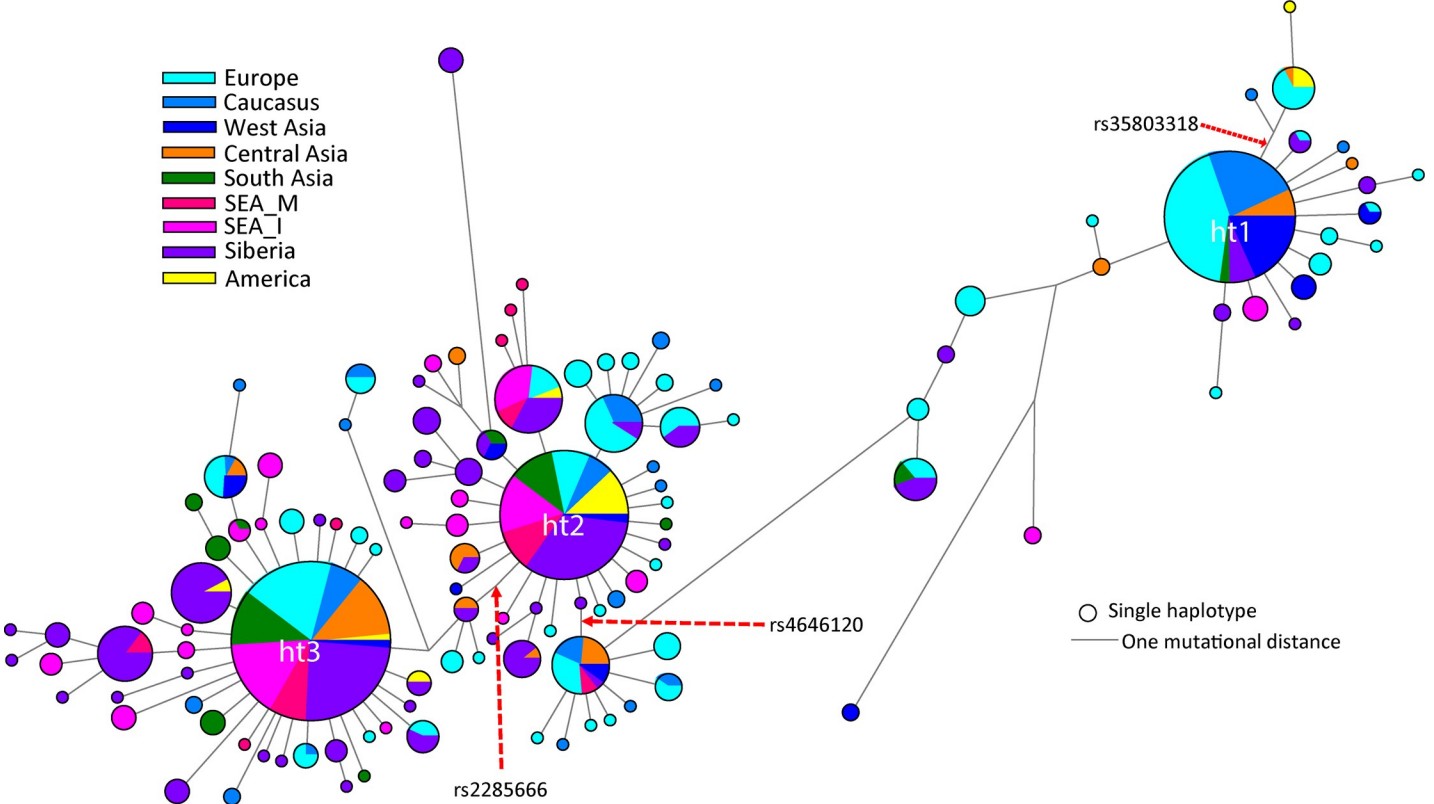

**Fig 2. The median joining (MJ) network of 142 haplotypes belonging to gene ACE2.** Circle sizes are proportional to the number of samples with that haplotype. The three most common haplotypes are marked. The three important SNPs studied in details have been marked in the figure. We used median joining method implemented in the NETWORK programme ver. 5.

SNPs play a key role in the distinction between East and West Eurasian populations (Fig 2 and S3 and S4 Figs). Interestingly, the most frequent haplotypes of South Asia involve these SNPs. A recent study has also highlighted the highest frequency of this SNP (rs 2285666) among Chinese populations (0.5) as well as significant frequency differences among 1000 genome populations (S4 Fig) [4]. In our study, we also found high frequency (0.6) of this SNP among South Asians (S2 Table and S4 Fig). Moreover, we also found that a synonymous coding region variant rs35803318 was most frequent among Americans (0.15), followed by Europeans (0.055), Caucasians (0.051) and Central Asians (0.021), whilst this site was not polymorphic for West Asians, South Asians, Southeast Asians and Siberians (S2 Table).

Phylogenetic analysis has suggested that the majority of South Asian samples share with East Eurasians the monophyletic haplotypes 2 and 3 by the unique polymorphism events (rs4646120) and (rs2285666). Recent studies have suggested that the reference allele has a reduced *ACE2* expression of up to 50%, resulting in greater severity of a SARS-CoV-2 infection [23–25]. Additionally, a synonymous coding region variant rs35803318 was also significantly more polymorphic among Americans and Europeans than among South Asians. Hence, it is likely that among South Asians, the host susceptibility to the novel coronavirus SARS-CoV-2 more closely resembles that of East/Southeast Asians rather than that of Europeans or Americans.

## Supporting information

**S1 Fig. The LD (linkage disequilibrium) plots of ACE2 gene of various studied populations.** Shading from white to red indicates the intensity of $r^2$ from 0 to 1. Strong LD is represented by a high percentage (>80) and a darker red square.
(TIF)

**S2 Fig. Heat map showing the intra- and inter-population variation measured by average pairwise sequence differences of the ACE2 gene among 1000 genome populations.** The average pairwise differences between populations are shown in the upper triangle of the matrix (green). The average number of pairwise differences within each population group are shown along the diagonal (orange). The differences between populations based on Nei's genetic distances are depicted in lower triangle of the matrix (blue). The populations are grouped in to superpopulations e.g. European, South Asian, East Asian and American.
(TIF)

**S3 Fig. The median joining (MJ) network of 491 haplotypes belonging to gene ACE2 among 1000 genome populations.** Circle sizes are proportional to the number of samples with that haplotype. The three most common haplotypes are marked. All three SNPs studied in detail have been marked by arrow. We used median joining method implemented in the NETWORK programme ver. 5.
(TIF)

**S4 Fig. The spatial distribution of alleles of rs4646120 and rs2285666 among 1000 genome populations.** The map was obtained from the *PGG* toolkit implemented in the https://www.pggsnv.org/.
(TIF)

**S1 Table. The number of samples from each of the region used in the analysis.** The number of South Asian groups shown with their linguistic affiliations.
(PDF)

**S2 Table. The details of 248 polymorphic loci extracted from analysed data.** The frequencies of alternate alleles for each loci and group have been mentioned in the table. The three important SNPs studied in details have been highlighted.
(XLSX)

## Acknowledgments

We thank to both of the reviewers and the Editor for their constructive suggestions.

## Author Contributions

**Conceptualization:** Rakesh Tamang, Gyaneshwer Chaubey.

**Data curation:** Rudra Kumar Pandey, Prajjval Pratap Singh, Pramod Kumar, Avinash Arvind Rasalkar, Gyaneshwer Chaubey.

**Formal analysis:** Anshika Srivastava, Rudra Kumar Pandey, Pankaj Shrivastava, Gyaneshwer Chaubey.

**Investigation:** Pramod Kumar, Rakesh Tamang, Gyaneshwer Chaubey.

**Project administration:** Gyaneshwer Chaubey.

**Supervision:** Gyaneshwer Chaubey.

**Validation:** Rudra Kumar Pandey, Prajjval Pratap Singh.

**Writing – original draft:** Anshika Srivastava, Gyaneshwer Chaubey.

**Writing – review & editing:** Prajjval Pratap Singh, Pramod Kumar, George van Driem.

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
