## [Decision Letter · Decision Letter 0]

2 Jun 2020

PONE-D-20-12219

Most frequent South Asian haplotypes of ACE2 share identity by descent with East Eurasian populations

PLOS ONE

Dear Dr. Chaubey,

Thank you for submitting your manuscript to PLOS ONE. After careful consideration, we feel that it has merit but does not fully meet PLOS ONE’s publication criteria as it currently stands. Therefore, we invite you to submit a revised version of the manuscript that addresses the points raised during the review process.

In particular, the Reviewer 1 raised major points about dataset and analyses, pointing out even problems of reproducibility and suggesting a "comparison to 1000G data", and about english writing, "typos but also sentences, issues with references and figures", which should be revised by a native speaker.

Please address also the other issues raised by both Reviewers and submit your revised manuscript by the 30th June 30, 2020. If you will need more time than this to complete your revisions, please reply to this message or contact the journal office at plosone@plos.org. Please include the following items when submitting your revised manuscript:

We look forward to receiving your revised manuscript.

Kind regards,

Alessandro Achilli, Ph.D.

Academic Editor

PLOS ONE

Journal Requirements:

This work is supported by the National Geographic Explorer grant HJ3-182R-18. The

funders had no role in study design, data collection and analysis, decision to publish or

preparation of the manuscript.

The authors received no specific funding for this work.

The authors have declared that no competing interests exist.

We note that one or more of the authors are employed by a commercial company: Redcliffe Life Sciences Pvt Ltd.

5. Please upload a copy of Supporting Information Table 2 which you refer to in your text on page 10.

Reviewers' comments:

Reviewer's Responses to Questions

**Comments to the Author**

1. Is the manuscript technically sound, and do the data support the conclusions?

Reviewer #1: Partly

Reviewer #2: Yes

2. Has the statistical analysis been performed appropriately and rigorously? 

Reviewer #1: N/A

Reviewer #2: Yes

3. Have the authors made all data underlying the findings in their manuscript fully available?

Reviewer #1: No

Reviewer #2: Yes

4. Is the manuscript presented in an intelligible fashion and written in standard English?

Reviewer #1: No

Reviewer #2: Yes

5. Review Comments to the Author

Reviewer #1: Srivastava et al present “Most frequent South Asian haplotypes of ACE2 share identity by descent with East Eurasian populations”. While the basic concept is interesting and timely given the current SARS-CoV-2 pandemic, the manuscript unfortunately can not be published in its current form:

- Unfortunately important information is missing, reproducibility is not entirely given

- the manuscript was written quickly and includes several errors, which make it hard to read but also trust the results, e.g. wrong SNP name.

- Unfortunately the images in Figure1 where hardly readable, with the figure legends not very informative and helpful

Major Issues:

The main issue I see with this manuscript is that it was written in a hustle, which a reader can easily tell, as there is information missing, one would typically expect, besides several errors (typos but also sentences, issues with references and figures). Comparison to 1000G data is very limited; as the main focus is on South Asia, with only 25 samples included, whereas 1000G includes 662 samples. Therefore one of my main concerns is, how representative are the 25 (mostly males, as ACE2 on X?) samples for South Asia. How do the results differ to the one reported by Cao et al 2020, what is the frequency of the two main SNPs in South-Asians for 1000G data?

The file in the provided data includes 402 samples, Pagani et al describe 483 samples. This work according the haplotypes in Suppl.Materials a total of 393 samples. Can you describe the discordance?

Unfortunately (technical issue here) the figures can not be read well – can you please elaborate and include a larger font? Also the figure legends are not very informative.

Please check the manuscript with help of a native english speaker

Minor Issues:

Abstract:

- include sample size of your data set, including for super populations once mentioned (EAS, SAS, EUR).

- the human Angiotensin-converting enzyme 2 (ACE2)

- more related TO West Eurasians

-focusing ON South Asia

-The second SNP – without talking about the first SNP? Why only the second one, what is with the first one?

Introduction:

- Please recheck sentence: ...has been one of the serious threat to humanity now

- encoded by A gene located ON

- expression of ACE2 receptor? relate to the severity OF THE disease

- Confusing sentence: Growing evidence are accumulating, however, still the expression of ACE2 variations affecting host susceptibility among various world population is not known.

- South Asia is a country ...really? It is a region consisting of countries Afghanistan, Bangladesh, Bhutan, Maldives, Nepal, India, Pakistan and Sri Lanka 

- Please recheck, very confusing: For the ACE2, it is not known that how the polymorphisms present in South Asia is shared with the East and West Eurasian populations?

-Reference 10 and 11 are identical, Reference 3 missing journal,

Material and Methods:

- Reference 10 vs 11. a [11] (Supplementary Table 1) here I was searching the data in Paganis Supplementary table – please make clearer

- Supplementary Figure 3: one can clearly see that the frequency of unique polymorphisms is strongly correletad to the sample size with one exception (SEA_M, this gets obvious when you order by sample size) – what is the purpose of the barplot? Same with Figure 1, how useful is this plot, can the sample size difference also be biasing the main finding?

- What is the purpose of supplemental table 1 and figure 3 – as figure 3 reports the same data?

- Please check sentence: … were constructed from the Haploview.

ResultS and discussion

- Please check: Europeans and Siberians had THE highest number of private polymorphisms which ARE likely due to their large number of samples from these groups (Supplementary Figure 1, Figure 3, Table 1)

- what are the snps defining ht1?

- LD patters → LD patterns

- this unexpected result – wasn’t it reported by Cao et al 2020 yet?

- LD Plots – why are different amount of SNPs used for the subfigures on population level? However no chance to read. Can you please provide a better description of how the LD

- consistent naming of haplotypes ht1 … either with space or none

- rs446120 is on chr8 – should be rs4646120 according figure 1c. 2X wrong in text.

- was more common in West Eurasians

- What leads to the conclusion that rs2285666 has lower severity for SARS-CoV-2?

Reviewer #2: Comments on the manuscript entitled “Most frequent South Asian haplotypes of ACE2 share identity by descent with East Eurasian populations”, by Srivastava et al.

Srivastava et al. have analysed whole genome data of the populations around the world, focusing South Asia, and extracted 248 ACE2 variations. Phylogenetic analysis of the above data suggested that the majority of South Asian ACE2 haplotypes are closer to the East Eurasians in the background of two unique event polymorphisms. Hence, the authors have suggested that it is highly likely that among the South Asians, the host susceptibility to the novel coronavirus SARS-CoV-2 will be more similar to East/Southeast Asians rather than the Europeans.

Considering the ongoing pandemic, this manuscript is appropriate now. However, I suggest the authors to take care of the following points.

1. No author has affiliation No. 5!

2. No mention about the total number of samples, analysed; although supplementary Table 1 has the information. I suggest the authors to mention the numbers in the methods.

3. It may be useful to discuss in the manuscript about the linguistic background of South Asian populations that were included in the study.

6. PLOS authors have the option to publish the peer review history of their article (what does this mean?). If published, this will include your full peer review and any attached files.

Reviewer #1: No

Reviewer #2: No

---

## [Author Response · Author response to Decision Letter 0]

30 Jun 2020

Reviewers' comments:

Reviewer's Responses to Questions

Comments to the Author

1. Is the manuscript technically sound, and do the data support the conclusions?

Reviewer #1: Partly

Reviewer #2: Yes

2. Has the statistical analysis been performed appropriately and rigorously?

Reviewer #1: N/A

Reviewer #2: Yes

3. Have the authors made all data underlying the findings in their manuscript fully available?

Reviewer #1: No

Reviewer #2: Yes

4. Is the manuscript presented in an intelligible fashion and written in standard English?

Reviewer #1: No

Reviewer #2: Yes

5. Review Comments to the Author

Reviewer #1: Srivastava et al present “Most frequent South Asian haplotypes of ACE2 share identity by descent with East Eurasian populations”. While the basic concept is interesting and timely given the current SARS-CoV-2 pandemic, the manuscript unfortunately can not be published in its current form:

We have substantially revised the manuscript by addressing all the concerns of the reviewers. 

- Unfortunately important information is missing, reproducibility is not entirely given

- the manuscript was written quickly and includes several errors, which make it hard to read but also trust the results, e.g. wrong SNP name.

- Unfortunately the images in Figure1 where hardly readable, with the figure legends not very informative and helpful

We have separated the figures now in the revised version.

Major Issues:

The main issue I see with this manuscript is that it was written in a hustle, which a reader can easily tell, as there is information missing, one would typically expect, besides several errors (typos but also sentences, issues with references and figures). Comparison to 1000G data is very limited; as the main focus is on South Asia, with only 25 samples included, whereas 1000G includes 662 samples. Therefore one of my main concerns is, how representative are the 25 (mostly males, as ACE2 on X?) samples for South Asia. How do the results differ to the one reported by Cao et al 2020, what is the frequency of the two main SNPs in South-Asians for 1000G data?

We have replicated our analysis with the 1000G data phase 3 data now in the revised version (Supplementary Figs 2-4). Results from both of the data has yielded similar results and conclusions drawn by us is supported by the analysis of independent datasets (Pagani et al. 2016 and 1000G). 

We differ from Cao et al 2020 paper substantially. Cao et al have just looked the frequency of coding region substitutions among 1000G as well as Chinese dataset. They also didn’t add any analysis by the ancestry sharing point of view. Whereas, we used more robust haplotype based approach and moved a step ahead than Cao et al. 

To have a better understanding of these SNPs worldwide, in the revised version, we have added the spatial distribution of two main SNPs in 1000G data (Supplementary Fig 4).

The file in the provided data includes 402 samples, Pagani et al describe 483 samples. This work according the haplotypes in Suppl.Materials a total of 393 samples. Can you describe the discordance?

According to the Supplementary Table 1. Pagani used 447 samples for their diversity panel. We have used the same data. After excluding regions e.g. Africa and Sahul as well as close relatives, we landed to 393 samples. We have explicitly mentioned this in the material and method section. 

Unfortunately (technical issue here) the figures can not be read well – can you please elaborate and include a larger font? Also the figure legends are not very informative.

We have made separate figures now and also improved the legends.

Please check the manuscript with help of a native english speaker

We have included a co-author who has improved the quality of writing as well as edited the manuscript for the language.

Minor Issues:

Abstract:

- include sample size of your data set, including for super populations once mentioned (EAS, SAS, EUR).

- the human Angiotensin-converting enzyme 2 (ACE2)

- more related TO West Eurasians

-focusing ON South Asia

We have revised the manuscript as per the suggestions of the reviewer.

-The second SNP – without talking about the first SNP? Why only the second one, what is with the first one?

We have expanded the discussion about both of the SNPs in the main text.

Introduction:

- Please recheck sentence: ...has been one of the serious threat to humanity now

- encoded by A gene located ON

- expression of ACE2 receptor? relate to the severity OF THE disease

We have revised the sentences accordingly.

- Confusing sentence: Growing evidence are accumulating, however, still the expression of ACE2 variations affecting host susceptibility among various world population is not known.

We have rephrased the sentence.

- South Asia is a country ...really? It is a region consisting of countries Afghanistan, Bangladesh, Bhutan, Maldives, Nepal, India, Pakistan and Sri Lanka 

We are sorry for this error! We have corrected it in the revised text.

- Please recheck, very confusing: For the ACE2, it is not known that how the polymorphisms present in South Asia is shared with the East and West Eurasian populations?

The sentence is revised now.

-Reference 10 and 11 are identical, Reference 3 missing journal,

We have rechecked all the references for the consistency.

Material and Methods:

- Reference 10 vs 11. a [11] (Supplementary Table 1) here I was searching the data in Paganis Supplementary table – please make clearer

These errors have been corrected.

- Supplementary Figure 3: one can clearly see that the frequency of unique polymorphisms is strongly correletad to the sample size with one exception (SEA_M, this gets obvious when you order by sample size) – what is the purpose of the barplot? Same with Figure 1, how useful is this plot, can the sample size difference also be biasing the main finding?

We agree with the reviewer and excluded these figures. To verify the main finding we used 1000G data as well as 26 random chromosomes for each of the groups. We didn’t get any deviation from the main finding in these analyses.

- What is the purpose of supplemental table 1 and figure 3 – as figure 3 reports the same data?

We agree with the reviewer. Since in the revised version we excluded Supp Fig. 3, we retained the Supplementary Table 1.

- Please check sentence: … were constructed from the Haploview.

It has been corrected now.

ResultS and discussion

- Please check: Europeans and Siberians had THE highest number of private polymorphisms which ARE likely due to their large number of samples from these groups (Supplementary Figure 1, Figure 3, Table 1)

We have excluded the figures as well as the sentence.

- what are the snps defining ht1?

We have updated Supp Table 1 SNPs with their location in different haplotypes.

- LD patters → LD patterns

It has been removed.

- this unexpected result – wasn’t it reported by Cao et al 2020 yet?

Cao et al. have looked the coding SNPs and didn’t analyse ancestry sharing, whereas we looked haplotypes and their sharing.

- LD Plots – why are different amount of SNPs used for the subfigures on population level? However no chance to read. Can you please provide a better description of how the LD

The program Haploview only consider the polymorphic SNPs for LD reconstruction, the polymorphic SNPs vary in different regions, therefore, the different amount of SNPs.

- consistent naming of haplotypes ht1 … either with space or none

It has been consistent now.

- rs446120 is on chr8 – should be rs4646120 according figure 1c. 2X wrong in text.

We are sorry for this typo, it is corrected in the manuscript.

- was more common in West Eurasians

Revised accordingly.

- What leads to the conclusion that rs2285666 has lower severity for SARS-CoV-2?

Recent studies have suggested that the Reference allele has reduced the ACE2 expression upto 50%, resulting the higher severity for SARS-CoV-2 (Singh et al. 2020; Wu et al., 2017; Asselta et al., 2020)

Reviewer #2: Comments on the manuscript entitled “Most frequent South Asian haplotypes of ACE2 share identity by descent with East Eurasian populations”, by Srivastava et al.

Srivastava et al. have analysed whole genome data of the populations around the world, focusing South Asia, and extracted 248 ACE2 variations. Phylogenetic analysis of the above data suggested that the majority of South Asian ACE2 haplotypes are closer to the East Eurasians in the background of two unique event polymorphisms. Hence, the authors have suggested that it is highly likely that among the South Asians, the host susceptibility to the novel coronavirus SARS-CoV-2 will be more similar to East/Southeast Asians rather than the Europeans.

Considering the ongoing pandemic, this manuscript is appropriate now. However, I suggest the authors to take care of the following points.

Thanks for the good words.

1. No author has affiliation No. 5!

We are sorry for this error. We have corrected it in the revised manuscript.

2. No mention about the total number of samples, analysed; although supplementary Table 1 has the information. I suggest the authors to mention the numbers in the methods.

We have revised the manuscript to reflect these informations.

3. It may be useful to discuss in the manuscript about the linguistic background of South Asian populations that were included in the study.

Thanks. We have added a supplementary table (table S1) with the linguistic information of South Asian samples.

6. PLOS authors have the option to publish the peer review history of their article (what does this mean?). If published, this will include your full peer review and any attached files.

Do you want your identity to be public for this peer review? For information about this choice, including consent withdrawal, please see our Privacy Policy.

Reviewer #1: No

Reviewer #2: No

We hope that in the revised version you will find everything in order.

Sincerely, 

G Chaubey (on behalf of coauthors)

---

## [Decision Letter · Decision Letter 1]

4 Aug 2020

PONE-D-20-12219R1

Most frequent South Asian haplotypes of ACE2 share identity by descent with East Eurasian populations

PLOS ONE

Dear Dr. Chaubey,

Thank you for submitting your manuscript to PLOS ONE. After careful consideration, we feel that it has merit but does not fully meet PLOS ONE’s publication criteria as it currently stands. Therefore, we invite you to submit a revised version of the manuscript that addresses the few minor points raised by the Reviewer 1.

We look forward to receiving your revised manuscript.

Kind regards,

Alessandro Achilli, Ph.D.

Academic Editor

PLOS ONE

Reviewers' comments:

Reviewer's Responses to Questions

**Comments to the Author**

1. If the authors have adequately addressed your comments raised in a previous round of review and you feel that this manuscript is now acceptable for publication, you may indicate that here to bypass the “Comments to the Author” section, enter your conflict of interest statement in the “Confidential to Editor” section, and submit your "Accept" recommendation.

Reviewer #1: All comments have been addressed

Reviewer #2: All comments have been addressed

2. Is the manuscript technically sound, and do the data support the conclusions?

Reviewer #1: Yes

Reviewer #2: Yes

3. Has the statistical analysis been performed appropriately and rigorously? 

Reviewer #1: Yes

Reviewer #2: Yes

4. Have the authors made all data underlying the findings in their manuscript fully available?

Reviewer #1: Yes

Reviewer #2: Yes

5. Is the manuscript presented in an intelligible fashion and written in standard English?

Reviewer #1: Yes

Reviewer #2: Yes

6. Review Comments to the Author

Reviewer #1: I thank the authors for addressing my concerns and for the additional work carried out. The manuscript is now almost ready for publication - only few minor issue:

In the abstract there are 383 samples while you mentioned 393 in material and methods (which I can replicate now)

S1 Table - as we are on Chromosome X - Chromosomes (2n) describes the data incorrectly, as there are 258 males and only 135 females - therefore suggest to write number of individuals

S2 Table - highlight the three SNPs described in the main text - would prefer an Excel file over the PDF file here.

Figure S3: can you please indicate the SNP rs3503318 in Figure 2 and S3?

Figure 1b and S2: can you please specify the axis values on the legends for diversity and describe in the figure legend respectively?

Reviewer #2: I have gone through the revised manuscript and the pint-by-point reply to my comments. I am satisfied with all the reply to my previous questions. I have no more question.

7. PLOS authors have the option to publish the peer review history of their article (what does this mean?). If published, this will include your full peer review and any attached files.

Reviewer #1: No

Reviewer #2: No

---

## [Author Response · Author response to Decision Letter 1]

5 Aug 2020

To,

Dr. Alessandro Achilli,

Academic Editor

PLOS ONE

Date: 05/08/2020

Dear Dr Achilli,

We have revised our manuscript considering the comments from reviewer1. Our pointwise answers are given below (in bold letters). 

///////////////////////////////////////////////////////////////////////////////////////////////////////////////

Reviewer #1: I thank the authors for addressing my concerns and for the additional work carried out. 

We are grateful to reviewers for their constructive suggestions which has helped us to improve our manuscript substantially. 

The manuscript is now almost ready for publication - only few minor issue:

In the abstract there are 383 samples while you mentioned 393 in material and methods (which I can replicate now)

We have updated the number accordingly. 

S1 Table - as we are on Chromosome X - Chromosomes (2n) describes the data incorrectly, as there are 258 males and only 135 females - therefore suggest to write number of individuals

Thanks for the great suggestion. We have corrected the table accordingly. 

S2 Table - highlight the three SNPs described in the main text - would prefer an Excel file over the PDF file here.

We have followed the suggestions and updated the manuscript accordingly. 

Figure S3: can you please indicate the SNP rs3503318 in Figure 2 and S3?

We have indicated the SNP in both of the figures. 

Figure 1b and S2: can you please specify the axis values on the legends for diversity and describe in the figure legend respectively?

The values have been already mentioned in the each of the color scale. 

We hope that you will find everything in order.

Sincerely,

G. Chaubey (on behalf of co-authors)

---

## [Editor Report · Decision Letter 2]

13 Aug 2020

Most frequent South Asian haplotypes of ACE2 share identity by descent with East Eurasian populations

PONE-D-20-12219R2

Dear Dr. Chaubey,

We’re pleased to inform you that your manuscript has been judged scientifically suitable for publication and will be formally accepted for publication once it meets all outstanding technical requirements.

Kind regards,

Alessandro Achilli, Ph.D.

Academic Editor

PLOS ONE

---

## [Editor Report · Acceptance letter]

19 Aug 2020

PONE-D-20-12219R2 

Most frequent South Asian haplotypes of ACE2 share identity by descent with East Eurasian populations 

Dear Dr. Chaubey:

I'm pleased to inform you that your manuscript has been deemed suitable for publication in PLOS ONE. Congratulations! Your manuscript is now with our production department. 

Kind regards, 

on behalf of

Prof. Alessandro Achilli 

Academic Editor

PLOS ONE